# The ETS Homologous Factor (EHF) Represents a Useful Immunohistochemical Marker for Predicting Prostate Cancer Metastasis

**DOI:** 10.3390/diagnostics12040800

**Published:** 2022-03-24

**Authors:** Manuel Scimeca, Manuela Montanaro, Rita Bonfiglio, Lucia Anemona, Enrico Finazzi Agrò, Anastasios D. Asimakopoulos, Roberto Bei, Vittorio Manzari, Nicoletta Urbano, Erica Giacobbi, Francesca Servadei, Elena Bonanno, Orazio Schillaci, Alessandro Mauriello

**Affiliations:** 1Department of Experimental Medicine, University of Rome “Tor Vergata”, Via Montpellier 1, 00133 Rome, Italy; manuela.montanaro@uniroma2.it (M.M.); rita.bonfiglio@uniroma2.it (R.B.); anemona@uniroma2.it (L.A.); erica.giacobbi@ptvonline.it (E.G.); francesca.servadei@ptvonline.it (F.S.); elena.bonanno@uniroma2.it (E.B.); alessandro.mauriello@uniroma2.it (A.M.); 2San Raffaele University, Via di Val Cannuta 247, 00166 Rome, Italy; 3Faculty of Medicine, Saint Camillus International University of Health Sciences, Via di Sant’Alessandro 8, 00131 Rome, Italy; 4Department of Surgical Sciences, Division of Urology, University of Rome Tor Vergata, 00133 Rome, Italy; finazzi.agro@med.uniroma2.it (E.F.A.); tasospao2003@yahoo.com (A.D.A.); 5Department of Clinical Sciences and Translational Medicine, University of Rome “Tor Vergata”, 00133 Rome, Italy; bei@med.uniroma2.it (R.B.); manzari@med.uniroma2.it (V.M.); 6Nuclear Medicine Unit, Department of Oncohaematology, Policlinico “Tor Vergata”, Viale Oxford 81, 00133 Rome, Italy; n.urbano@virgilio.it; 7Department of Biomedicine and Prevention, University of Rome “Tor Vergata”, Via Montpellier 1, 00133 Rome, Italy; orazio.schillaci@uniroma2.it; 8Istituto di Ricovero e Cura a Carattere Scientifico (IRCCS) Neuromed, Via Atinense, 18, 86077 Pozzilli, Italy

**Keywords:** prostate cancer, ETS homologous factor, metastasis, histological marker

## Abstract

The main aim of this study was to investigate the risk of prostate cancer metastasis formation associated with the expression of ETS homologous factor (EHF) in a cohort of bioptic samples. To this end, the expression of EHF was evaluated in a cohort of 152 prostate biopsies including primary prostate cancers that developed metastatic lesions, primary prostate cancers that did not develop metastasis, and benign lesions. Data here reported EHF as a candidate immunohistochemical prognostic biomarker for prostate cancer metastasis formation regardless of the Gleason scoring system. Indeed, our data clearly show that primary lesions with EHF positive cells ≥40% had a great risk of developing metastasis within five years from the first diagnosis. Patients with these lesions had about a 40-fold increased risk of developing metastasis as compared with patients with prostate lesions characterized by a percentage of EHF positive cells ≤30%. In conclusion, the immunohistochemical evaluation of EHF could significantly improve the management of prostate cancer patients by optimizing the diagnostic and therapeutic health procedures and, more important, ameliorating the patient’s quality of life.

## 1. Introduction

Prostate cancer is the male neoplasia with the highest prevalence and the second incidence in worldwide [1]. About 80% of prostate cancers are localized at the first diagnosis, whilst 20% have spread from primary mass to regional lymph nodes (13%) or to distant organs (6–7%) [1]. The presence of metastasis at the first diagnosis significantly impacts the 5-year survival rate. Indeed, the 5-year survival rate is about 100% for prostate cancer patients with localized mass and only 30% for those with evidence of tumour metastasis. Therefore, the occurrence of metastatic lesions represents a major health challenge in the management of prostate cancer patients, despite the slow growth rate of prostate lesions; metastatic lesions usually develop after 10 years from the diagnosis.

At the state of the art, the main prognostic factors of the prostate cancers are the pre-treatment Gleason score [2], the clinical stage [3], the PSA level [4], and the percentage of positive core biopsy [5]. However, some of these lack specificity, e.g., PSA level, that is subject to both aging and presence of inflammation [5]. As concerns the Gleason score, the International Society of Urological Pathology released supplementary guidance in 2014 and a revised prostate cancer grading system called the Grade Groups in order to improve the prognostic and predictive values of this morphological grading system [2].

Nevertheless, the mortality rate, although decreasing, remains high [1], probably due to cancer resistance after treatment. This suggests that prostate cancers with similar morphological appearance may significantly differ from a biological or molecular point of view. A better stratification of prostate lesions with high risk of developing metastasis based on the combination of morphological and molecular characteristics of tumours could provide fundamental information for choosing the most appropriate therapeutic approach, thus improving the management of prostate cancer patients. In particular, the identification of easy-to-investigate prognostic biomarkers, such as immunohistochemical biomarkers, could improve the capability of the Gleason scoring system to find patients with high risk of metastasis development [6].

Moreover, the research of biomarkers capable of stratifying prostate cancer patients according to the risk of developing metastatic lesions could also improve knowledge about the molecular mechanisms related to prostate cancer progression.

Several studies proposed mechanisms and biomarkers associated with the capability of prostate cancer cells to invade the surrounding tissues and thus to form metastatic lesions [7,8,9,10,11]. The loss of epithelial characteristics, i.e., the epithelial to mesenchymal transition (EMT) phenomenon, seems to be very effective in promoting prostate tumour invasion [12,13,14,15]. The expression of EMT inductor molecules from the tumour environment, as well as the acquisition of a mesenchymal phenotype by prostate cancer cells are considered negative prognostic biomarkers for cancer progression [16,17].

One of the most interesting pathways related to the EMT phenomenon involves the ZEB family proteins, such as ZEB1/δEF1 and ZEB2/SIP1, which are crucial modulators of the EMT phenomenon [16,17,18]. In this scenario, ETS homologous factor (EHF), a molecule of the epithelium-specific subfamily (E26 transformation-specific (ETS) transcription factor family), seems to be involved in the regulation of the EMT by activating the ZEB1 promoters [19,20]; however, the involvement of EHF in cancer occurrence and progression is a very debated issue in the oncological research. Indeed, EHF takes part in the complex molecular network implicated in the modulation of epithelial cell differentiation and stem-like potential under physiological conditions [21,22]. Sakamoto and colleagues [23] recently demonstrated that different EHF transcript variants, i.e., EHF-LF and EHF-SF, may have different impact on the differentiation of epithelial cells, thus promoting or inhibiting the EMT phenomenon. Specifically, authors found that new point mutations within the conserved ETS domain of EHF increase the tumour invasion in vivo by abolishing the native function of EHF protein. This evidence could explain the apparently contradictory data concerning the role of EHF in cancer progression. In fact, some studies highlight the protective role of EHF in cancer development, underling its ability to preserve the epithelial characteristics of cells [24,25], whilst other studies emphasize the impact of EHF over-expression in EMT-related cancer metastatization [19,26]. The EHF expression was also investigated in prostate cancer. The main studies concerning the effect of EHF in prostate cancer cells have been performed by the Prostate Cancer Biology and Experimental Therapeutics groups of the Institute of Oncology Research [27,28,29,30,31]. Among these, Albino et al. [29] performed in vitro and in vivo investigations in which it has been demonstrated that EHF influences cancer progression by modulating the IL-6/JAK/STAT3 pathway in prostate cancer stem cells. In this model, EHF can be considered a molecular factor capable of inhibiting the expansion of prostate cancer stem cell compartments. However, to the best of our knowledge, no studies have been performed about the possible association between EHF expression and the formation of prostate cancer metastasis by investigating human biotic samples.

Therefore, starting from these considerations, the main aim of this study was to investigate the risk of prostate cancer metastasis formation associated with the expression of EHF in a cohort of bioptic samples.

## 2. Materials and Methods

### 2.1. Prostate Samples Collection

In this study 152 consecutive prostate samples were retrospectively collected from treatment-naïve patients (73.06 ± 2.53 years; range 64–85 years). At least 16 bioptic specimens were analysed for the histopathological diagnosis. For each patient, the sample with the highest value of Gleason group (GG) was used for immunohistochemical evaluation of EHF.

The main anamnestic data were registered, as well as the follow-up concerning the development of distant metastasis within 5-years of the histological diagnosis.

From each bioptic sample, paraffin serial sections were used for both histological and immunohistochemical investigations.

This retrospective study was approved by the Institutional Ethical Committee of the “Policlinico Tor Vergata” (reference number #129.18). Experimental procedures were performed in agreement with The Code of Ethics of the World Medical Association (Declaration of Helsinki). All patients signed an informed consent prior to surgical procedures.

Surgical samples were formalin-fixed, paraffin-embedded, and haematoxylin and eosin stained, as previously described [32]. For each cancer lesion, the GG was evaluated. Histological classification was blindly performed by two pathologists (L.A. and A.M.). The presence of metastases was detected by imaging diagnostic analyses, such as positron emission tomography and magnetic resonance imaging.

### 2.2. Immunohistochemistry

Immunohistochemical analyses were performed to study the expression of EHF in prostate lesions. Briefly, sections were treated with EDTA citrate pH 7.8 for 30 min at 95 °C for the antigen retrieval [33]. Sections were then incubated for 1 h at room temperature with the rabbit polyclonal anti-EHF antibody (1:200; ab272671, ABCAM, Cambridge, CB2 0AX, UK). Washings were performed with PBS/Tween20 pH 7.6. Reactions were revealed by HRP-DAB Detection Kit (UCS Diagnostic, Rome, Italy). Immunoreaction was evaluated as percentage of EHF positive cells.

### 2.3. Bioinformatics Analysis

Bioinformatics analysis was carried out on the https://tnmplot.com/analysis/ website (accessed date 1 March 2022) by examining data generated by gene arrays from the Gene Expression Omnibus of the National Center for Biotechnology Information (NCBI-GEO) [34]. This analysis showed the results of EHF gene expression in “Normal” tissue, non-metastatic “Tumours”, and “Metastatic” tumours. 

### 2.4. Statistical Analysis

Data were analysed using SPSS version 16.0 (SPSS Inc., Chicago, IL, USA) software. Continuous variables were expressed as the mean ± SD or ± SE. The Shapiro–Wilk test was used to statistically assess the normal distribution of the data. Comparisons between continuous variables were performed using the Bonferroni post hoc test. Categorical data were analysed using the chi-square test or the Fisher exact test. Linear regression analysis was used to study the possible association between EHF expression and PSA values. In order to evaluate the odds ratio of metastases for different percentage of EHF positive tumour cells, different cut-offs of positivity were tested (30%, 40%, and 50%).

In order to evaluate the risk of metastatization of prostate cancer lesions, the PSA values were subdivided in two group using 10 ng/mL as the cut-off in which patients with PSA < 10 ng/mL had a low risk of disease progression [5]. Similarly, the values of the histological grading, evaluated in terms of GG, were subdivided into two groups. The first group included patients with GG1, generally considered to have a favourable prognosis [5], whilst the other ones included patients with GG ≥ 2.

The odds ratio of metastases for serum PSA, GG scores, and that for different percentage of EHF-positive cells was evaluated by logistic regression using the value of EXP (B), where B represents the logistic coefficient. A 2-tailed *p* value < 0.05 was considered statistically significant.

## 3. Results

### 3.1. Prostate Lesions Classification

In this study, 152 prostate biopsies were retrospectively collected. All biopsies were classified according to WHO 2016 [5]. In particular, histological analysis allowed the biopsies to be classified as follows: 112 prostate acinar adenocarcinomas and 40 benign prostate hyperplasia (BL), defined as a nonmalignant enlargement of the prostate because of overgrowth of the epithelium and fibromuscular tissue in the transition zone and peri-urethral area [5]. According to 5-year follow-up data, the 112 prostate adenocarcinomas were further subdivided in lesions that developed distant metastasis within 5 years from the first diagnosis (n = 15; PC+) and lesions that did not develop distant metastasis within 5 years from the first diagnosis (n = 97; PC−). The highest value of GG was used.


*According to the GG, lesions were classified as follows: 15.9% GG1 (Gleason score ≤ 6), 9.4% GG2 (Gleason score 3 + 4), 16.5% GG3 (Gleason score 4 + 3), 21.5% GG4 (Gleason score 4 + 4), and 8.1% GG5 (Gleason score ≥ 9). No specific association was observed between the GG and the occurrence of metastasis. In all patients who underwent prostatectomy during the follow-up (n = 12), histological analysis confirmed the GG value of the first diagnosis (i.e., the highest GG value obtained in the biopsies).*


### 3.2. EHF Expression

The expression of EHF was assessed by immunohistochemical analyses. For each tissue, the percentage of positive epithelial cells (both cancer and benign) was evaluated on the whole biopsy (Figure 1A–C).

As concerns EHF expression, a significant group effect was observed in PC+, PC−, and BL lesions (one-way ANOVA *p* < 0.0001) (Figure 1A). Post hoc test (Bonferroni comparisons test) revealed a significant increase in the percentage of EHF-positive prostate cancer cells in PC+ (62.9 ± 4.4%) as compared with both PC− (26.8 ± 2.3%) and BL (4.9 ± 1.1%) lesions (PC+ vs. PC− *p* < 0.0001; PC+ vs. BL *p* < 0.0001; PC− vs. BL *p* < 0.0001) (Figure 1A).

Of note is that no significant association was observed by studying the relationship between EHF expression and the GG (Figure 2A,B). Specifically, neither significant data distribution (One way ANOVA GG *p* = 0.548; GG *p* = 0.735) nor significant differences in post hoc analysis were observed.

### 3.3. Bioinformatics Analysis

The examination of data generated by NCBI-GEO (https://tnmplot.com/analysis/) (accessed date 1 March 2022) [34] further supported EHF immunohistochemical data. Indeed, gene chip data from NCBI-GEO showed a significant increase in EHF gene expression in prostate cancers (tumour n = 283) and metastatic cancers (Metastatic n = 6) as compared with prostate-normal tissues (normal n = 106) (*p* = 3.65 × 10^−1^) (Figure 2). Although the low number of metastatic prostate cancer data collected in NCBI-GEO dataset does not allow conclusive considerations, an increase in the EHF gene expression was also observed in the metastatic cancers if compared with tumour ones.

### 3.4. Risk of Prostate Metastasis Formation

Logistic regression analysis was applied to identify the risk of prostate cancer metastasis formation associated with the GG or the expression of EHF (Table 1(a)).

As concerns the GG, an increase in the risk of prostate metastasis formation was observed in the GG4 (1.41 95% CI 0.29–6.79) as compared with other GGs (Table 1). Specifically, GG4 showed an ~8-fold increase in the risk of prostate cancer metastatization with respect to low grade lesions (GG2 0.19 95% CI 0.02–2.04).

Three cut-offs of EHF-positive prostate cancer cell percentages were tested (30%, 40%, and 50%) (Table 1). Remarkably, the relative risk of prostate cancer metastasis formation was exceptionally low for patients with primary lesions characterized by 30% of EHF-positive prostate cancer cells but considerably increased for value of EHF expression ≥40% (Table 1). It is important to underline that over the 40% threshold, the expression of EHF was related to 40-fold increase in prostate metastasis development risk. Indeed, only 7.1% of prostate biopsies of PC+ group showed values of EHF-positive cells <40% (Table 1).

Noteworthy, multivariate analysis showed a great risk of prostate cancer metastasis formation in primary lesions with EHF expression >40%, regardless of both PSA and GG (Table 1(b)).

## 4. Discussion

Data here reported EHF as a candidate for an immunohistochemical prognostic biomarker for prostate cancer metastasis formation, regardless of the GG. Indeed, our data clearly show that primary lesions with a percentage of EHF-positive cells ≥40% had a great risk of developing metastasis within five years from the first diagnosis. Patients with these lesions have about a 40-fold increased risk of developing metastasis as compared with patients with prostate lesions characterized by a percentage of EHF-positive cells ≤30%.

The EHF is a transcription factor of the ETS family that plays a fundamental role in the molecular network that orchestrates gene expression in the epithelial cells in response to both endogenous and exogenous stimuli [35]. More generally, the ETS proteins are transcription factors that exert several biological functions of the epithelium thanks to the highly conserved DNA-binding domains [35]. As concerns human cancer, these transcription factors showed a paradoxical effect in that they can both inhibit and promote cancer progression [19,36]. Hence, EHF expression has also been associated with both tumour occurrence/progression and its inhibition. However, to the best of our knowledge, no studies have been performed about the risk of prostate cancer metastasis formation related to EHF expression.

To this end, in this study the expression of EHF was evaluated in a cohort of 152 prostate biopsies, including primary prostate cancers that developed metastatic lesions (PC+), primary prostate cancers that did not develop metastasis (PC−), and benign lesions (BL). Noteworthy, our data show a significant increase in the percentage of EHF-positive cells in PC+ if compared with both PC− and BL, thus suggesting a possible role of this molecule in prostate cancer progression. It is also interesting to note that no association between EHF expression and GG has been observed. This evidence allows speculation that EHF expression could be a prognostic factor regardless of GG, also suggesting that specific molecular signatures can explain the different metastatic potential of prostate cancer lesions with similar morphological aspects.

To further strengthen the immunohistochemical data here displayed, a bioinformatics analysis was carried out on the https://tnmplot.com/analysis/ (accessed date 1 March 2022) [34]. EHF gene expression showed the same trend as the immunohistochemical data, with an increase in EHF expression in prostate metastatic lesions with respect to both prostate cancers and benign lesions. The limit of this bioinformatics analysis, as well as for the immunohistochemical study, is the low number of metastatic lesions included in the case selection.

As reported above, logistic regression analysis associated the EHF expression in primary prostate cancer with a risk of developing metastatic lesions within 5 years from the diagnosis, establishing a threshold value of 40% of positive cells, beyond which the risk increases 40-fold. Indeed, just ~7% of prostate cancers that gave rise to metastatic lesions that showed a percentage of EHF positive cells ≤40%.

From a biological point of view, these data could be explained by both the presence of different EHF transcript variants, some of which were associated with EMT occurrence, and the accumulation of mutations capable of inducing the loss of function of the EHF protein; under physiological conditions, EHF participates in maintaining the epithelial characteristics of cells [23].

As concerns the EHF transcript variants, Sakamoto et al. [23] demonstrated that only EHF-SF variant abrogates ETS1-mediated activation of the ZEB1 promoter by promoting degradation of ETS1 proteins, thereby inhibiting the EMT phenotypes of cancer cells. Conversely, the EHF-LF variant could favour EMT-related cancer progression by stimulating ZEB1 promoter. According to this hypothesis, the biological link between EHF expression and the formation of prostate cancer metastasis may be represented by the EMT phenomenon. In fact, it is known that the EMT makes the epithelial cancer cells able to invade the surrounding tissues, thus forming distant metastasis [37]. However, it is important to note that the polyclonal antibody used in this study that recognized different epitopes of EHF did not discriminate different variants of this molecule. Thus, the observed expression accounts for all known variants of EHF.

Additionally, data from our study can be explained by assuming the loss of EHF function due to the accumulation of mutations in its gene [23]. These mutations should be able of impairing EHF function but not its expression. Similar to what occurs for other molecules, it is possible to speculate that the EHF loss of function can activate feedback in which gene expression is strongly stimulated in an effort to re-establish physiological status. Under these conditions, the loss of the protective role of EHF can be associated with an increase in its expression, as observed in this study.

It is important to underline that these hypotheses must be confirmed through transcriptomics and molecular biology studies.

## 5. Conclusions

In the era of precision medicine, the discovery of new prognostic biomarkers capable of identifying prostate cancer lesions with high metastatic potential represents one of the main goals of oncological research. Indeed, the early identification of prognostic characteristics of tumours is essential for establishing better therapeutic approaches.

The results of this study highlight the predictive value of EHF, suggesting its possible use in the characterization of prostate cancer lesions. Specifically, the evaluation of EHF expression can significantly improve the management of prostate cancer patients, optimizing diagnostic and therapeutic health procedures, and more important, ameliorating the patient’s quality of life. Lastly, the possibility of evaluating EHF expression by easily available techniques, such as immunohistochemistry, allows this prognostic information to be obtained quickly during histopathological analyses.

## Figures and Tables

**Figure 1 diagnostics-12-00800-f001:**
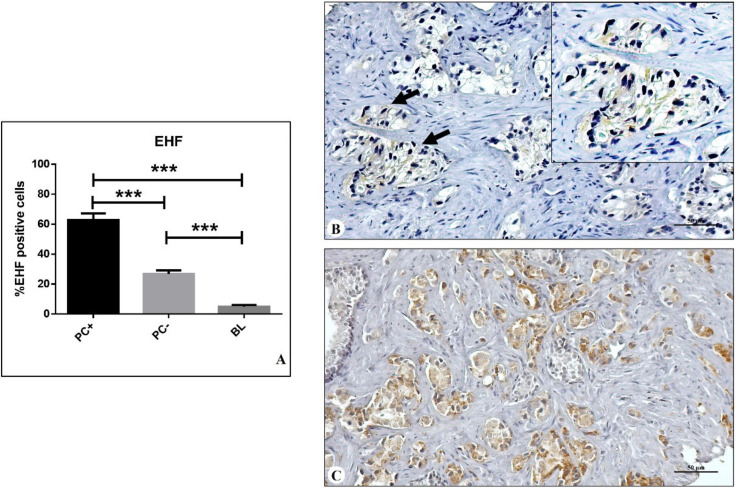
Immunohistochemical evaluation of EHF. (**A**) Graph shows the percentage of EHF-positive cells in prostate lesions that developed metastasis within 5 years of the first diagnosis (PC+), prostate lesions that did not develope metastasis within 5 years from the first diagnosis (PC−), and benign lesions (BL). (**B**) PC− lesion with EHF-positive cells ≤30% (arrows). (**C**) PC+ lesion with EHF-positive cells ≥30%. *** *p* < 0.001.

**Figure 2 diagnostics-12-00800-f002:**
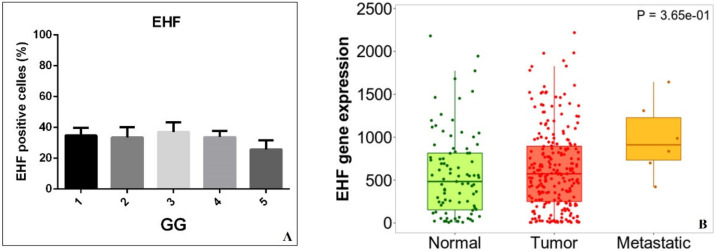
EHF expression, Gleason group (GG), and bioinformatics analysis. (**A**) The graph displays the percentage of EHF-positive cells in prostate cancer lesions classified according to the Gleason group (GG). (**B**) EHF gene expression in normal tissue, prostate adenocarcinomas, and prostate cancers that developed metastatic lesions (metastatic).

**Table 1 diagnostics-12-00800-t001:** SPA, Gleason group, EHF expression, and the risk of prostate metastasis formation.

a				
	ODDs RATIO	95% CI		*p*
		Lower	Upper	
EHF30	0.381	0.296	0.491	0.001
EHF40	45.043	5.62	361.27	0.001
EHF50	42.000	8.49	207.86	0.001
**b**				
	**ODDs RATIO**	**95% CI**		** *p* **
		**Lower**	**Upper**	
PSA *	1.40	0.29	6.77	0.67
GG **	7.31	0.72	74.48	0.09
EHF ***	50.19	5.90	427.12	0.001

* PSA > 10 ng/mL, ** GG ≥ 2, *** EHF ≥ 40%.

## Data Availability

Data will be provided on request.

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
