# Peer review of "The ETS Homologous Factor (EHF) Represents a Useful Immunohistochemical Marker for Predicting Prostate Cancer Metastasis"

_diagnostics, 2022, doi:10.3390/diagnostics12040800_

Round 1

Reviewer 1 Report

In the present study, the authors have immunohistochemically assessed the expression of EHF in prostate cancer specimens and its association with the risk of metastasis. They thus deal with an interesting topic. However, the following need to be considered or adequately addressed.

1) “40” and “30” (lines 28/30/181/182/216/219/251) should represent percentages, but not the numbers of positive cells.

2) The present results do not readily indicate the clinical impact of EHF expression “in combination with” other prognostic factors (lines 31/282). Further analysis with combined data is required to state this.

3) It is unclear how Gleason score/Grade Group was defined in each patient when different parts of the biopsy specimen showed different scores. Is it the highest in each case? In addition, how was Gleason score determined if, for example, one core from right apex showed 3+3 and another from the same site showed 4+4?

4) It is unclear how many cores/specimens in each case were stained. If stained in only one core or specimen, how was it selected?

5) Percentage of EHF positive cells:

a) It is unclear if only epithelial (i.e. cancer, benign) cells vs. both epithelial and stromal cells were evaluated.

b) The entire biopsy specimens, instead of 5 representative HPFs, should be assessed.

6) 25.5% Gleason score 7 vs. 9.4% (GG2) + 16.5% (GG3) = 25.9%; 29.9% Gleason score 8 or higher vs. 21.5% (GG4) + 8.1% (GG5) = 29.6%

7) Grade Groups are simple translation of Gleason scores. Analyses of both (e.g. Fig. 2A vs. 2B) appear to be redundant and that of Grade Groups only may suffice.

8) It is often difficult and may not be difficult to determine if the glands are and stromal tissue is, respectively, hyperplastic on biopsies. Do the 40 BL cases primarily represent stomal hyperplasia? If so, how was the immunoreactivity determined (also see Comment 5-a)?

9) Fig. 2B: No definitive signals are seen near the arrows. Do the authors intend to show a “negative” case?

10) Fig. 2C data: It is unclear if the “Tumor” includes the cases with metastasis. If such information is available, then the expression levels can be compared between those with vs. without metastasis.

11) Table 1 data:

a) P values appear to be required.

b) For the analysis of Grade Groups, does GG1 represent a reference (i.e. odds ratio 1)?

c) What does “0.381” indicate? Is the risk 62% lower in cases with EHF =>30% than in those with EHF <30%?

12) Title: EHF expression does not represent a “histologic” marker.

13) Other minor points:

a) A total of 112 prostate cancer patients, particularly only 15 who have developed metastasis, may not represent a “large” cohort (lines 22/103).

b) There are no “Gleason groups” (lines 31/61/193/196/200/212/282).

c) It is unclear if both lymph node and distant metastases are defined as “metastasis” in this study.

d) Why were 5 “adjacent” HPFs (line 131) examined?

e) References 9/33 and 19/37 are identical, respectively.

Author Response

Before we begin the point-by-point review of the list of concerns, we would like to thank the Reviewer for their comments on how to improve the manuscript, which has been revised accordingly, as well as the Editors for calling for a new submission of an improved version of our manuscript.

Reviewer#1

In the present study, the authors have immunohistochemically assessed the expression of EHF in prostate cancer specimens and its association with the risk of metastasis. They thus deal with an interesting topic. However, the following need to be considered or adequately addressed.

Reply: we would like to thank the Reviewer for expressing interest in our work, and for their availability to review our manuscript.

  • “40” and “30” (lines 28/30/181/182/216/219/251) should represent percentages, but not the numbers of positive cells.

Reply: Thanks for this point out. We corrected the manuscript accordingly.

2) The present results do not readily indicate the clinical impact of EHF expression “in combination with” other prognostic factors (lines 31/282). Further analysis with combined data is required to state this.

Reply: thanks for this point out. In the new version of the manuscript, we deleted the sentences in which the EHF expression was related to “other prognostic factors”

3) It is unclear how Gleason score/Grade Group was defined in each patient when different parts of the biopsy specimen showed different scores. Is it the highest in each case? In addition, how was Gleason score determined if, for example, one core from right apex showed 3+3 and another from the same site showed 4+4?

Reply: Thanks for this point out. For each patient, it was collected in the study the specimen with the highest value of Grade Group.  In the new version of the manuscript, we added this info.

4) It is unclear how many cores/specimens in each case were stained. If stained in only one core or specimen, how was it selected?

Reply: thanks for this point out. As reported above, for each patient, we collected the specimen with the highest value of Grade Group.

5) Percentage of EHF positive cells:

  1. a) It is unclear if only epithelial (i.e. cancer, benign) cells vs. both epithelial and stromal cells were evaluated.
  2. b) The entire biopsy specimens, instead of 5 representative HPFs, should be assessed.

Reply: thanks for this point out.

  1. Our analysis was focused only on epithelial cells (both cancers and benign). We specified this in the methods paragraph.
  2. We re-evaluate the whole biopsy specimens but not differences were noted.

6) 25.5% Gleason score 7 vs. 9.4% (GG2) + 16.5% (GG3) = 25.9%; 29.9% Gleason score 8 or higher vs. 21.5% (GG4) + 8.1% (GG5) = 29.6%

Reply: thanks for this point out. We emended these typos.  

7) Grade Groups are simple translation of Gleason scores. Analyses of both (e.g. Fig. 2A vs. 2B) appear to be redundant and that of Grade Groups only may suffice.

Reply: thanks for this point out. We agree with the reviewer about the possible redundancy of the data showed in the figure 2 (Grade group and Gleason score). Thus, we deleted the results concerning the Gleason score. 

8) It is often difficult and may not be difficult to determine if the glands are and stromal tissue is, respectively, hyperplastic on biopsies. Do the 40 BL cases primarily represent stomal hyperplasia? If so, how was the immunoreactivity determined (also see Comment 5-a)?

Reply: thanks for this point out. As reported in the Result paragraph, histological diagnosis was performed according to the WHO 2016. In the new version of the manuscript, we better explain the diagnostic procedures also highlight the number of bioptic specimens investigated. This allows us to also identify the benign lesions. 

Specifically, the text was modified as follow:

Methods paragraph

In this study 152 consecutive prostate samples were retrospectively collected from treatment-naïve patients (73.06 ± 2.53 years; range 64–85 years At least 16 bioptic specimens were analyzed for the histopathological diagnosis. For each patient, the sample with the highest value of Gleason Group was used for immunohistochemical evaluation of EHF.

Results paragraph

In this study 152 prostate biopsies were retrospectively collected. All biopsies were classified according to WHO 2016 [5]. In particular, histological analysis allowed to classify them as follow:  112 prostate acinar adenocarcinomas and 40 prostate hyperplasia (BL). According to 5 years follow-up data, the 112 prostate adenocarcinomas were further subdivided in lesions that developed distant metastasis within 5 years from the first diagnosis (n=15 PC+) and lesions that have not developed distant metastasis within 5 years from the first diagnosis (n=97; PC-). The highest value of GG has been used.

9) Fig. 2B: No definitive signals are seen near the arrows. Do the authors intend to show a “negative” case?

Reply: we provided with more details the staining in the figure 2B.

10) Fig. 2C data: It is unclear if the “Tumor” includes the cases with metastasis. If such information is available, then the expression levels can be compared between those with vs. without metastasis.

Reply: thanks for this point out. The “tumour” group includes cases without metastases. We specified this.

Method paragraph

This analysis showed the results of EHF gene expression in “Normal” tissue, non-metastatic “Tumours” and “Metastatic” ones. 

11) Table 1 data:

  1. a) P values appear to be required.

  1. b) For the analysis of Grade Groups, does GG1 represent a reference (i.e. odds ratio 1)?

  1. c) What does “0.381” indicate? Is the risk 62% lower in cases with EHF =>30% than in those with EHF <30%?

Reply: thanks for this point out. We added the p value in the table 1. Odds ratio for GG1 was unvaluable due to the low number of patients characterized by a Gleason score ≤6 (according to the new analysis – multivariate analysis – we modified the table 1). 0.381 represents the odds ratio. As reported in the discussion paragraph the odds ratio values indicate that “primary lesions with a percentage of EHF positive cells ≥40% have a great risk to develop metastasis within five years from the first diagnosis. Patients with these lesions have about a 40-fold increased risk to develop metastasis as compared with patients with prostate lesions characterized by a percentage of EHF positive cells ≤30%.”

12) Title: EHF expression does not represent a “histologic” marker.

Reply: we modified the title as follow:

The ETS homologous factor (EHF) represents a useful immunohistochemical marker to predict prostate cancer metastasis

13) Other minor points:

  1. a) A total of 112 prostate cancer patients, particularly only 15 who have developed metastasis, may not represent a “large” cohort (lines 22/103).

Reply: we deleted the word “large” in highlighted sentences.

  1. There are no “Gleason groups” (lines 31/61/193/196/200/212/282).

Reply: thanks for this point out.

  1. It is unclear if both lymph node and distant metastases are defined as “metastasis” in this study.

Reply: only distant metastases were considered. We added this information in the new version of the study.

  1. Why were 5 “adjacent” HPFs (line 131) examined?

Reply: we re-observed all biopsies evaluating the whole sample. No differences where observed. We corrected the manuscript accordingly.

  1. e) References 9/33 and 19/37 are identical, respectively.

Reply: we corrected the references.

Reviewer 2 Report

Scimeca et al. found that higher EHF expression with a high risk to develop prostate cancer metastasis within five years from the first diagnosis and regardless of Gleason grade. According to the authors' retrospective study of 152 consecutive prostate biopsies statistical analysis, they claim EHF has useful histological prognostic value. Regarding biomarker qualification of EHF in clinical prostate cancer use, there are still many points to be clarified and provided evidence is not convincing enough. The author needs to provide more study to help the integrity of the paper is necessary. Some suggestions are as follows:

Major suggestions:

  1. Is the EHF an independent prognostic factor? It’s important to provide the univariate and multivariate Cox regression analysis of EHF and other clinical features (PSA level, pathological stage, Grade group, and recurrence) in the PC+ and PC- groups.
  2. Does PC+ vs PC- in 5 years follow-up data show a difference in survival?
  3. EHF expression without difference between the Gleason score or Grade group at the start time point of study shown in figure 2A and 2B; what is the expression of EHF in the follow-up between grade groups after 5 years?
  4. Were these specimens obtained from treatment-naïve cancer? If patients received treatments, the correlation of treatment and EHF in a cohort of 152 prostate biopsies should be addressed and sub-grouping into different treatments, that will provide more strategies and applications for the treatment of prostate cancer metastasis.
  5. As author hypothesizes that EHF seems to be involved in EMT progression, however the connection between the EHF and metastasis is insufficiency. Authors are encouraged to do transwell migration or wound-healing assay of knockdown or overexpression of EHF in PCa cell to validate whether EHF indeed affects metastasis. In addition, the ZEB family and EMT genes also need to be validated under EHF knockdown in PCa to provide a suitable mechanism.

Minor suggestions:

  1. In the introduction, lines 70-72 need to cite the references for the argument.
  2. As the authors mention in the Introduction lines 82-83, which variant of EHF do the authors believe (EHF-LF or truncated EHF) contribute to prostate cancer metastasis in the study?
  3. In figure 2A, the X-axis is not clearly marked and does not match the legend.
  4. Typos in line 95 "the L-6/JAK/STAT3"
  5. Table 1 all Odds ratios are comma not dot; EHF30 Odds ratio value is incomplete.

Author Response

Before we begin the point-by-point review of the list of concerns, we would like to thank the Reviewer for their comments on how to improve the manuscript, which has been revised accordingly, as well as the Editors for calling for a new submission of an improved version of our manuscript.

Reviewer#2

Scimeca et al. found that higher EHF expression with a high risk to develop prostate cancer metastasis within five years from the first diagnosis and regardless of Gleason grade. According to the authors' retrospective study of 152 consecutive prostate biopsies statistical analysis, they claim EHF has useful histological prognostic value. Regarding biomarker qualification of EHF in clinical prostate cancer use, there are still many points to be clarified and provided evidence is not convincing enough. The author needs to provide more study to help the integrity of the paper is necessary.

Reply:  we would like to thank the Reviewer for expressing interest in our work, and for their availability to review our manuscript.

Is the EHF an independent prognostic factor? It’s important to provide the univariate and multivariate Cox regression analysis of EHF and other clinical features (PSA level, pathological stage, Grade group, and recurrence) in the PC+ and PC- groups.

Reply: Thanks for this point out. In the new version of the manuscript, we added a multivariate analysis in which we analyzed the GG, the PSA value and the EHF expression. Data of this analysis further emphasize the possible prognostic value of EHF expression. We added this data in the new version of the manuscript. In particular, we added a new table 1.

Does PC+ vs PC- in 5 years follow-up data show a difference in survival?

Reply: 5 years follow-up data showed no difference in survival. Indeed, all patients were alive after 5-yeasr.

EHF expression without difference between the Gleason score or Grade group at the start time point of study shown in figure 2A and 2B; what is the expression of EHF in the follow-up between grade groups after 5 years?

Reply: In all patients underwent to prostatectomy during the follow-up (n=12), histological analysis confirmed the GG value of the first diagnosis (i.e. the highest GG value obtained in the biopsies). Similarly, in these 12 specimens no significant differences in the EHF expression were noted. However, in our opinion, the low cases with prostatectomy during the follow-up do not allow solid consideration about the EHF expression.

Were these specimens obtained from treatment-naïve cancer? If patients received treatments, the correlation of treatment and EHF in a cohort of 152 prostate biopsies should be addressed and sub-grouping into different treatments, that will provide more strategies and applications for the treatment of prostate cancer metastasis.

Reply: all specimens were obtained from treatment-naïve cancer. We specified this in the new version of the manuscript.

As author hypothesizes that EHF seems to be involved in EMT progression, however the connection between the EHF and metastasis is insufficiency. Authors are encouraged to do transwell migration or wound-healing assay of knockdown or overexpression of EHF in PCa cell to validate whether EHF indeed affects metastasis. In addition, the ZEB family and EMT genes also need to be validated under EHF knockdown in PCa to provide a suitable mechanism.

Reply: we thank the reviewer for proposing this experimental idea to us. We will certainly continue our studies concerning the effect of EHF expression in prostate cancer. However, according to the aim of the journal, in this study we focused our attention on the possible prognostic impact of EHF. The molecular mechanisms underline our evidences have only been hypothesized.

As the authors mention in the Introduction lines 82-83, which variant of EHF do the authors believe (EHF-LF or truncated EHF) contribute to prostate cancer metastasis in the study?

Reply: According to the paper of Sakamoto et al. “Only EHF-SF abrogates ETS1-mediated activation of the ZEB1 promoter by promoting degradation of ETS1 proteins, thereby inhibiting the EMT phenotypes of cancer cells.” Therefore, EHF-LF could be involved in the EMT phenomenon.

We modified the discussion as follow:

As concern the EHF transcript variants, Sakamoto et al. [23] demonstrated that only EHF-SF variant abrogates ETS1-mediated activation of the ZEB1 promoter by promoting degradation of ETS1 proteins, thereby inhibiting the EMT phenotypes of cancer cells. Conversely, the EHF-LF variant could favors the EMT related cancer progression by stimulating ZEB1 promoter.

in the introduction, lines 70-72 need to cite the references for the argument.

In figure 2A, the X-axis is not clearly marked and does not match the legend.

Typos in line 95 "the L-6/JAK/STAT3"

Table 1 all Odds ratios are comma not dot; EHF30 Odds ratio value is incomplete.

Reply: we corrected these errors.

Round 2

Reviewer 1 Report

The authors have adequately addressed some of the concerns from this reviewer. However, the following still need to be considered. The numbers correspond to the original ones.

1) “a percentage of EHF positive cells XX%” (Abstract, legends for Fig. 1) should be rephrased.

2) Line 32: What is “in situ” evaluation?

8) Again, do all 40 benign tissues represent “hyperplasia” (line 168), but not simply “benign” or “non-neoplastic”?

9) This reviewer is sorry, but the comment was for Fig. 1B. The arrows do not appear to be helpful.

10) This reviewer is unable to find the changes stated in the authors’ response. Also, “metastatic prostate cancers” (line 217) may be confusing (expression in primary tumors in cases with metastasis vs. metastatic sites).

14) New minor points:

a) GG is defined twice.

b) Sections 2.1. and 2.2. can be combined. Also, some of the sentences appear to be redundant or too general.

c) Similar to the above comment (#2), “in-situ” (e.g., lines 267, 271, 293) may be confusing, since this is an immunohistochemical study, but not in-situ hybridization.

Author Response

Before we begin the point-by-point review of the list of concerns, we would like to thank the Reviewer for their comments on how to improve the manuscript, which has been revised accordingly, as well as the Editors for calling for a new submission of an improved version of our manuscript.

Reviewer#1

The authors have adequately addressed some of the concerns from this reviewer. However, the following still need to be considered.

Reply: we would like to thank the Reviewer for expressing interest in our work, and for their availability to review our manuscript.

1) “a percentage of EHF positive cells XX%” (Abstract, legends for Fig. 1) should be rephrased. Reply: Thanks for this point out. We corrected the manuscript accordingly.

2) Line 32: What is “in situ” evaluation?

Reply: thanks for this point out. Immunohistochemistry is considered an in-situ techniques. However, according to the reviewer suggestion we substituted the term in situ with the immunohistochemical in all the manuscript.

8) Again, do all 40 benign tissues represent “hyperplasia” (line 168), but not simply “benign” or “non-neoplastic”?

Reply: all the “hyperplasia” included in this study have been classified according to the WHO. To better specify this, we modified the manuscript as follow:

Result paragraph

Histological analysis allowed to classify them as follow:  112 prostate acinar adenocarcinomas and 40 benign prostate hyperplasia (BL) defined as a nonmalignant enlargement of the prostate because of overgrowth of the epithelium and fibromuscular tissue of the transition zone and peri-urethral area [5].

9) This reviewer is sorry, but the comment was for Fig. 1B. The arrows do not appear to be helpful.

Reply: thanks for this point out. We modified the fig.1 accordingly.

10) This reviewer is unable to find the changes stated in the authors’ response. Also, “metastatic prostate cancers” (line 217) may be confusing (expression in primary tumors in cases with metastasis vs. metastatic sites).

Reply: we better specify this in the manuscript.

Legend figure 2

  1. B) EHF gene expression in normal tissue, prostate adenocarcinomas and prostate cancers that developed metastatic lesions (metastatic).

  1. a) GG is defined twice.

  1. b) Sections 2.1. and 2.2. can be combined. Also, some of the sentences appear to be redundant or too general.

  1. c) Similar to the above comment (#2), “in-situ” (e.g., lines 267, 271, 293) may be confusing, since this is an immunohistochemical study, but not in-situ hybridization.

Reply: we modified the text accordingly. 

Reviewer 2 Report

The author kindly responded to my questions one by one, revised as suggested, and provided relevant references as supporting materials.

Surprisingly, the patient who was diagnosed with prostate cancer did not receive further treatment, even in high GG.

Just a few minor issues left to solve:

  1. Except for 12 patients who underwent prostatectomy, how about EHF expression in other patients between grade groups after 5 years?
  2. Still typos in line 95 "the L-6/JAK/STAT3" needs to be corrected “IL-6/JAK/STAT3”

Author Response

Before we begin the point-by-point review of the list of concerns, we would like to thank the Reviewer for their comments on how to improve the manuscript, which has been revised accordingly, as well as the Editors for calling for a new submission of an improved version of our manuscript.

Reviewer#2

The author kindly responded to my questions one by one, revised as suggested, and provided relevant references as supporting materials.

Surprisingly, the patient who was diagnosed with prostate cancer did not receive further treatment, even in high GG.

Reply: we would like to thank the Reviewer for expressing interest in our work, and for their availability to review our manuscript.

As concern the treatment of patients is not true that patients did not receive any treatment after biopsy. In the text, we specified that patients did not receive treatment before biopsy. During the follow-up patients with GG1 generally did not receive any treatment. Conversely, patients with GG≥2 generally underwent to the common treatment modalities such as radiotherapy and hormones.

Method

In this study 152 consecutive prostate biopsies samples were retrospectively collected from treatment-naïve patients.

This was referred to the patients’ status before bioptic procedures (not during the follow-up).

Except for 12 patients who underwent prostatectomy, how about EHF expression in other patients between grade groups after 5 years?

Reply: we can evaluate the EHF expression only on bioptic samples. Therefore, the expression of EHF in other patients could not be assessed.

Still typos in line 95 "the L-6/JAK/STAT3" needs to be corrected “IL-6/JAK/STAT3”

Reply: we corrected this.